# Disseminated intravascular coagulation, associated factors and clinical outcomes among critically Ill septic adults admitted to a tertiary hospital in Ethiopia: A prospective longitudinal study

**Girum Tesfaye Kiya**[1]*, **Zeleke Mekonnen**[1], **Elsah Tegene Asefa**[2], **Gedion Milkias**[1], **Edosa Tadasa**[1], **Edosa Kejela**[3], **Iyasu Demeke**[1], **Aragaw Fiseha**[1], **Gemeda Abebe**[1]

1 School of Medical Laboratory Sciences, Jimma University, Jimma, Ethiopia, 2 Department of Internal Medicine, Jimma University, Jimma, Ethiopia, 3 Department of Anesthesia, Jimma University, Jimma, Ethiopia

* tesfaye.girum@ju.edu.et

## Abstract

### Background

Despite the established link between sepsis and disseminated intravascular coagulation (DIC), data on the prevalence of DIC, associated factors and patient outcomes in sepsis patients are inadequate in resource-limited settings. Therefore, the present study aimed to determine the magnitude of DIC and associated factors and mortality and predictors in septic adults admitted to intensive care units (ICUs).

### Methods

A prospective longitudinal study involving adults admitted to intensive care units was conducted. A structured checklist and questionnaire were used to collect patient demographic and clinical data. Blood samples were collected on days 1, 3, and 5 of admission for all laboratory analyses. A DIC diagnosis was made on the basis of the Japanese Association for Acute Medicine (JAAM) score. Descriptive statistics, multivariable logistic regression analysis, receiver operating characteristic (ROC) curve analysis and Kaplan–Meier survival analysis were employed in this study.

### Results

The overall prevalence of DIC in sepsis patients was 38 (25.7%). There were 24 (16.2%) patients who developed DIC on day 1 of admission, while 20 (19.4%) and 9 (12.9%) patients developed DIC on day 3 and day 5 of admission, respectively. Increased aspartate transaminase (AST) (AOR: 4.39; 95% CI: 1.75–11.01), thrombocytopenia (AOR: 6.04; 95% CI: 2.41–15.12), and prolonged prothrombin time

**Data availability statement:** All relevant data are within the manuscript and its Supporting Information files.

**Funding:** This study is supported by the Research and Innovation Director (RID) office of Institute of Health, Jimma University with Mega Research Fund Scheme (2022-2025). The funders had no role in the conceptualization, design, data collection and analysis, decision to publish, or preparation of the manuscript.

**Competing interests:** The authors have declared that no competing interests exist.

(PT) (AOR: 3.40; 95% CI: 1.36–8.51) were independent predictors of DIC in sepsis patients. There was no statistically significant difference in survival between patients with and without DIC (p < 0.328). The JAAM score at ICU admission predicted ICU mortality (AUC: 0.787; 95% CI: 0.624–0.950).

## Conclusion

A quarter of ICU-admitted septic adults developed DIC. The incidence was notably greater by the third day after admission, highlighting the importance of closely monitoring these patients for DIC progression. Elevated AST liver enzyme levels, thrombocytopenia, and prolonged PT are linked to the development of DIC. Changes in these variables could prompt further examination for DIC. The mortality rate did not significantly differ between septic patients with and without DIC. The JAAM score used to diagnose DIC in sepsis patients can serve as a predictor of ICU mortality in sepsis patients with DIC.

---

## Background

Disseminated intravascular coagulation (DIC) is a severe condition in which coagulation processes become excessive and uncontrollably activated throughout the body, posing a significant threat to life [1]. According to the International Society on Thrombosis and Hemostasis (ISTH) scientific subcommittee, DIC is defined as "an acquired syndrome characterized by the intravascular activation of coagulation with loss of localization arising from different causes. It can arise from the microvasculature and cause damage to it, which, if sufficiently severe, can cause organ dysfunction" [2].

The signature feature of DIC is the loss of localized activation of coagulation and the inefficiency of natural coagulation inhibitors in downregulating thrombin generation [1]. It also leads to the depletion of clotting factors and platelets, causing severe and potentially fatal bleeding [3]. The derangement of the fibrinolytic system further contributes to intravascular clot formation, but in some cases, accelerated fibrinolysis may cause severe bleeding [4]. DIC is not a disease in itself but rather a syndrome that always develops secondary to critical conditions [5], among which sepsis is the most frequent cause [6].

DIC is common in sepsis patients and is associated with poor prognosis in these patients [6,7]. In sepsis-associated DIC, the main event is a systemic inflammatory response to the infectious agent [8]. In sepsis, endothelial injury and subsequent tissue injury due to circulatory abnormalities cause multiorgan failure, and the ensuing DIC is a thromboinflammatory response that affects patient outcomes [9].

The activation of coagulation in sepsis is meant to protect the host against infection, which is detrimental to the host in cases of overactivation [10]. Crosstalk between inflammation and coagulation is considered crucial in the pathogenesis of sepsis, whereby inflammatory cytokines such as interleukin (IL)-1, IL-6 and tumor necrosis factor-α (TNFα) induce the activation of coagulation, whereas protease-activated receptors (PARs) are activated by coagulation proteases to initiate signaling

and subsequent cytokine production [11]. Tissue factor expression [12,13] during severe sepsis and the activation of platelets [14] are also key initiators of thrombin formation in sepsis.

Approximately one-third to half of sepsis patients admitted to the ICU are reported to have DIC [6,15]. The incidence of DIC in patients with sepsis ranges from 29%−61% [16,17]. Septic patients with DIC have a higher mortality rate than do patients without DIC [18]. In a study involving ICU patients with severe sepsis, the overall mortality rate was 21.5%. The mortality rate was 17.5% for non-DIC patients and 24.8% for those with DIC complications [17]. In many types of critically ill patients, the presence of DIC is an independent and relatively strong predictor of organ dysfunction and mortality [19] However, data regarding the magnitude of DIC in sepsis patients in developing countries are scarce.

The development of DIC in septic patients complicates their clinical management, as it leads to both thrombotic and hemorrhagic complications due to the consumption of clotting factors and platelets [20]. Early detection of coagulation disorders is crucial for assessing the severity and predicting the prognosis of sepsis [21]. Sepsis-induced DIC causes the development of microvascular and macrovascular thrombosis and increases the risk of bleeding due to consumptive coagulopathy [22]. In DIC, blockage caused by thrombin in microvascular and macrovascular tissues results in tissue ischemia, which can exacerbate organ dysfunction [22]. The development of multiple organ dysfunction syndrome (MODS) is a major determinant of mortality in sepsis patients [23]. Sepsis-induced DIC is associated with a 2-fold increase in mortality [24].

The burden of DIC among ICU-admitted sepsis patients is not well understood in resource-limited areas. Systematic and contextual data are essential for implementing effective management alternatives. Despite the established link between sepsis and DIC, data on the prevalence of DIC and its importance in determining patient outcomes, particularly in sepsis patients, are scarce. There is limited understanding of the specific risk factors that exacerbate DIC in septic patients admitted to the ICU. Therefore, the present study aimed to determine the magnitude of DIC and associated factors and ICU mortality and predictors in adult patients with sepsis.

## Methods

### Study setting, period and participants

A prospective longitudinal study involving adults admitted to intensive care units from October 1, 2023, to September 30, 2024, was conducted at Jimma University Medical Center (JUMC). JUMC is one of the oldest hospitals in Ethiopia, and it is the only teaching and referral hospital in southwest Ethiopia with 800 beds and a catchment population of over 20 million people. JUMC has a total of 12 beds in medical, emergency and surgical ICUs, with an estimated annual admission of 300 patients.

### Eligibility, sample size and sampling technique

All adult septic patients admitted to the ICUs of JUMC during the data collection period and willing to participate in the study were included. Patients who were receiving anticoagulant therapy before participation or with known clotting disorders, patients who were unable to provide consent themselves and patients whose consent could not be obtained from the next of kin were excluded from the study.

Accordingly, a total of 285 consecutive ICU admission episodes were screened for sepsis during a one-year period of data collection. Out of 285 ICU admissions, 148 patients were diagnosed with sepsis at the time of admission and were consecutively enrolled in this study.

### Demographic and clinical data collection

A structured checklist and questionnaire prepared in English and then translated to the Amharic and Afaan Oromoo versions were used for face–to–face interviews on sociodemographic data. Clinical data were obtained from patients'

medical records after informed consent was obtained and recorded until day five after admission or until the patient was discharged from the ICU if it took place earlier.

## Blood specimen collection, processing, and analysis

Blood samples were collected on days 1, 3, and 5 of admission via a citrated tube and an ethylene diamine tetra-acetic acid (EDTA) tube for coagulation analysis, complete blood count (CBC), and D-dimer analysis. The platelet (PLT) and white blood cell (WBC) counts were determined as part of the complete blood count (CBC) via an electrical impedance method-based hematology analyzer (Mindray BC-30 s) from an EDTA-anticoagulated blood sample. Platelet-poor plasma (PPP) was prepared from the citrated tube sample after centrifugation at 1500 RPM for 15 minutes to analyze the pro-thrombin time (PT), international normalized ratio of prothrombin time (PT/INR) and activated partial thromboplastin time (aPTT) via the URIT-610 coagulation analyzer. D-dimer was also analyzed by the Finecare III Plus analyzer using whole blood.

## Diagnosis of sepsis

The surviving sepsis campaign guideline recommends hospitals to select the most accurate and timely approach to sepsis screening that they can feasibly implement [25]. Accordingly, the diagnosis of sepsis was made by an attending physician based on clinical judgment by incorporating clinical presentations, laboratory findings, and radiological evidence. Clinical investigations included pulse rate, respiratory rate, temperature, oxygen saturation, and Glasgow Coma Scale, while the laboratory analyses included CBC, procalcitonin, C-reactive protein, and organ function tests. Culture-positive sepsis (proven sepsis) was identified in 15 adult patients whose clinical and laboratory findings were consistent with sepsis and in whom a causative pathogen was confirmed. Clinical sepsis was diagnosed in an additional 133-adult patients who exhibited clinical and laboratory features consistent with sepsis, but for whom causative organism was not identified or could not be demonstrated.

## Diagnosis of DIC

A DIC diagnosis was made on the basis of the JAAM DIC diagnostic criteria. The JAAM DIC score was calculated from clinical data and laboratory results obtained on days 1, 3, and 5 of ICU admission or until the patient was discharged from the ICU if it occurred earlier. A JAAM score ≥4 supports a diagnosis of DIC. The parameters utilized by the JAAM DIC criteria for subsequent analyses include systemic inflammatory response syndrome (SIRS), PT-INR, D-dimer, and the PLT [26]. All patients were followed until discharge from the ICU after enrollment in the study. The mortality of the patients was recorded during their stay in the ICUs. For detailed information on the JAAM DIC diagnostic criteria, see the supplementary data (S1 Table).

## Data analysis procedures

The data were checked for completeness and consistency, entered into Epidata version 3.1 (Epidata Association, Odense, Denmark) and exported to STATA version 17 for analysis. The normality of the data distribution was evaluated via the Shapiro–Wilk test. All continuous variables are presented as medians with interquartile ranges (IQRs), and categorical variables are expressed as frequencies (percentages). The patients were divided into 2 groups: those with sepsis with DIC and those without DIC. Differences between the groups were compared with the Mann–Whitney U test for continuous variables and with either the chi–square test or Fisher's exact test, as necessary for categorical variables. A binary logistic regression model was used to select a candidate variable for multivariate logistic regression analysis and to avoid confounding factors. All variables with a p value < 0.25 were selected for a multivariable logistic regression model. Multivariable logistic regression analysis was used to explore the predictors of DIC in sepsis patients. Receiver operating

characteristic (ROC) curve analysis was performed to determine the performance of the JAAM score in predicting mortality using the area under the curve (AUC). Kaplan–Meier survival analysis was run to determine the survival probability of septic patients with and without DIC.

## Quality assurance process

**Data quality assurance.** Before data collection, training was provided to the data collectors. Regular audits of the data collection processes were performed by the principal investigator to ensure the accuracy and reliability of the data. The integrity of the data was maintained by ensuring that the data were stored in a proper and organized manner to prevent loss and unauthorized access.

**Laboratory quality assurance.** Before blood sample collection, informed consent was obtained from the study participants after their purpose was explained. Labeling was performed on the sample container, the questionnaire paper and the request paper with the same identification number. The samples were checked for hemolysis and stored at the appropriate temperature if there was delay in analysis. Standard operational procedures (SOPs) were followed for each test. The quality of the automated hematology analyzer (Mindry, BC-30 s) was checked via duplicate tests on patient samples, carryover checks and background tests. The reagent package was examined for leakage, moisture absorption, and expiring date before use as per the manufacturer's instructions. The quality of the coagulation analyzer was ensured by control samples (AMP CoaTrol) and analyzed at the beginning of each day of work. Each laboratory test result was recorded and delivered to the physician. The test results were kept confidential.

## Ethical approval

This study was approved by the Jimma University Institute of Health Ethical Review Board with reference number JUIH/IRB/309/23 and followed the principles stated in the Declaration of Helsinki. The support letter was written by the research and innovation director office of Jimma University Institute of Health to JUMC, and permission was obtained from the medical director office of the medical center. Written informed consent was obtained from the study participants after the benefits and risks of the study were described, and those with impaired consciousness were enrolled with written consent from the next of kin. Any information concerning the participants was kept confidential, and the study participants had the right to withdraw from the study at any time. Patients were diagnosed with DIC, and their complete laboratory report was reported to the attending physician.

## Results

### Demographic and clinical characteristics of the study population

A total of 285 consecutive ICU admissions were registered, of whom 148 patients were diagnosed with sepsis at admission and were enrolled in the study. Among the enrolled patients, 103 (69.6%) reached the third day, and 70 (47.3%) reached the fifth day. This was because some patients get discharged or died in the ICUs. The enrollment flowchart is presented in Fig 1.

Among the septic patients, 82 (55.4%) were male, and the median age of the patients was 35 years. Most of the study participants were married (67.6%) or rural dwellers (69.6%). With respect to educational status, most of the study participants had no formal education (35.1%) and were at the primary educational level (37.8%). The most frequent site of infection was the lung (48.7%), and respiratory disease comorbidity affected 59 patients (39.9%). Similarly, lung failure was the leading type of organ failure, present in 55 individuals (37.2%). Among the 148 patients, 103 (69.6%) underwent mechanical ventilation, 94 (63.5%) received antibiotics, and 36 (24.3%) were treated with vasopressors. Fifteen patients (10.1%) had positive blood cultures. The median length of ICU stay of the study population was 3 days, with 36.5% in-ICU mortality. Comparisons of demographic and clinical characteristics between sepsis patients with and without DIC revealed

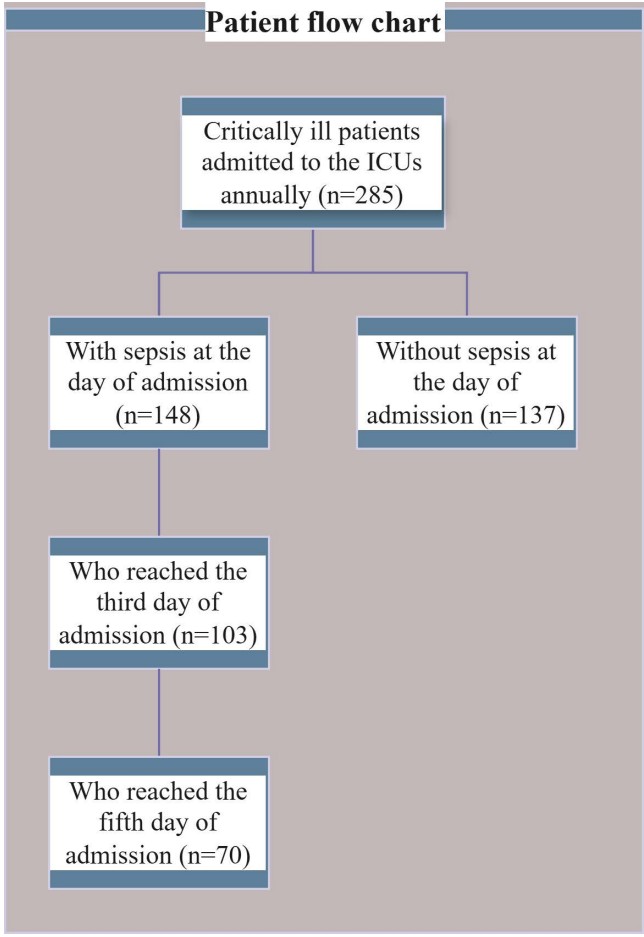

**Fig 1. Enrollment flowchart.**

no statistically significant differences between the groups. However, laboratory and clinical scores were significantly different between the two groups (Table 1).

**Prevalence of DIC in sepsis patients.** The overall prevalence of DIC in sepsis patients during the study period was 38/148 (25.7%). The number of patients who developed DIC on the day of admission was 24/148 (16.2%), whereas 20/103 (19.4%) and 9/70 (12.9%) patients developed DIC on day 3 and day 5 after admission, respectively, as shown in Fig 2. On the third day, 13 of the 20 DIC cases were newly developed, whereas on the fifth day, 4 of the 9 DIC cases were newly developed.

### Risk factors for disseminated intravascular coagulation

In the bivariate analysis, several variables, such as mechanical ventilation, respiratory comorbidities, malaria comorbidities, metabolic disease, positive blood culture, increased AST, increased ALT, and increased disease severity scores, such as the modified early score (MEWS), national early warning score (NEWS) and SIRS score, and laboratory parameters, such as thrombocytopenia, prolonged PT and prolonged APTT, were found to be significantly associated with DIC development. Analysis of all the candidate variables by multivariate analysis revealed that increased AST, thrombocytopenia, and prolonged PT were independent predictors of DIC in sepsis patients (Table 2).

**Table 1. Baseline and clinical characteristics of the study population according to the presence of DIC at JUMC from October 1, 2023, to September 30, 2024.**

| Variable | Overall (n = 148) | With DIC (n = 38) | Without DIC (n = 110) | p value |
|---|---|---|---|---|
| Age, years, median (IQR | 35 (25-45) | 30 (21.5-42) | 35 (25-45) | 0.168 |
| Male n (%) | 82 (55.4) | 20 (52.6) | 62 (56.4) | 0.690 |
| **Marital Status, n (%)** | | | | |
| Married | 100 (67.6) | 24 (63.2) | 76 (69.1) | 0.818 |
| Single | 44 (29.7) | 13 (34.2) | 31 (28.2)) | |
| Divorced | 3 (2) | 1 (2.6) | 2 (2) | |
| Widowed | 1 (0.7) | 0 (0) | 1 (0.7) | |
| **Residence, n (%)** | | | | |
| Rural | 103 (69.6) | 8 (21.1) | 37 (33.60 | 0.146 |
| Urban | 45 (30.4) | 30 (78.9) | 73 (66.4) | |
| **Education, n (%)** | | | | |
| No formal education | 52 (35.1) | 14 (36.8) | 38 (34.5) | 0.729 |
| Primary | 56 (37.8) | 16 (42.1) | 40 (36.4) | |
| Secondary | 19 (12.8) | 3 (7.9) | 16 (14.5) | |
| College and above | 21 (14.2) | 5 (13.2) | 16(14.5) | |
| **Site of infection, n (%)** | | | | |
| Respiratory tract | 72 (48.7) | 16(42.1) | 56 (51) | 0.325 |
| Abdomen | 26 (17.6) | 7 (18.4) | 19 (19.3) | 0.890 |
| Soft tissue | 11 (7.4) | 5 (13.2) | 6 (5.5) | 0.283 |
| CNS | 13 (8.8) | 4 (10.5) | 9 (8.2) | 0.742 |
| Other | 14 (9.5) | 5 (13.2) | 19 (17.3) | 0.355 |
| **Comorbidities, n (%)** | | | | |
| Respiratory disease | 59 (39.9) | 20 (52.6) | 39 (35.5) | 0.062 |
| Heart disease | 12 (7.4) | 2 (5.3) | 10 (9.10) | 0.731 |
| Malaria | 29 (19.6) | 11 (30) | 18 (16.4) | 0.177 |
| HTN | 8 (5.4) | 0 (0) | 8 (7.3) | 0.114 |
| DM | 14 (9.5) | 1 (2.6) | 13 (11.8) | 0.117 |
| Cancer | 5 (3.4) | 1 (2.6)) | 4 (3.6)) | 0.618 |
| **Organ failure, n (%)** | | | | |
| Liver failure | 5 (3.4) | 2 (5.3) | 3 (2.7) | 0.603 |
| Kidney failure | 15 (10.1) | 2 (5.3) | 13 (11.8) | 0.356 |
| Lung failure | 55 (37.2) | 16 (42.1) | 39 (35.5) | 0.465 |
| Brain failure | 16 (10.8) | 4 (10.5) | 12 (11) | 0.608 |
| **Number of organ failure, n (%)** | | | | |
| None | 51 (34.5) | 14 (36.8) | 37 (33.6) | 0.430 |
| One | 78 (52.7) | 22 (57.9) | 56 (51.9) | |
| Two | 17 (11.5) | 2 (5.3) | 15 (13.6) | |
| Three | 2 (1.4) | 0 (0) | 2 (1.8) | |
| **Interventions** | | | | |
| Mechanical ventilator | 103 (69.6) | 31 (81.6) | 72 (65.5) | 0.068 |
| Antibiotic use | 94 (63.5) | 22 (57.9) | 72 (65.50 | 0.511 |
| Vasopressor use | 36 (24.3) | 11 (28.9) | 25 (22.7) | 0.441 |
| Positive blood culture n (%) | 15 (10.1) | 6 (15.8) | 9 (8.2) | 0.180 |
| Length of ICU stay, median (IQR) | 3 (2-6.5) | 3 (1-5) | 3 (2-4.25) | 0.849 |

*(Continued)*

**Table 1.** (Continued)

| Variable | Overall (n = 148) | With DIC (n = 38) | Without DIC (n = 110) | p value |
|---|---|---|---|---|
| ICU mortality, n (%) | 54 (36.5) | 17 (44.7) | 37 (33.6) | 0.222 |
| PLT (x10⁹/L), median, IQR[1] | 178 (106-274) | 70(33.5-197.75) | 199.5 (139.75-297.75) | <0.001 |
| Thrombocytopenia n (%)[1] | 46 (31.1) | 26 (68.42) | 20 (18.20) | <0.001 |
| PT1 (sec), median, IQR[1] | 13.9 (12.4-16.5) | 15.6(13.60-18.50) | 13.6(12.10-16.40) | 0.006 |
| Prolonged PT, n (%)[1] | 54 (36.49) | 21(55.26) | 33 (30) | 0.005 |
| APTT1(sec), median, IQR[1] | 29.9 (26.4-37.9) | 36(28.80-44.80) | 28.650 (25.93 −36.53) | 0.010 |
| Prolonged APTT, n (%)[1] | 51 (34.50) | 20 (52.63) | 31(28.2) | 0.006 |
| UVA1 score, median, IQR[1] | 4 (0.5-5) | 2 (0–5) | 4 (1234–5) | 0.342 |
| UVA score>5, n (%)[1] | 49 (33.1) | 11 (28.9) | 38 (34.5) | 0.527 |
| MEWS score, median, IQR[1] | 5 (4-7) | 6 (5-8) | 5 (3-7) | 0.013 |
| MEWS score >5, n (%)[1] | 95 (64.2) | 31(81.6) | 64 (58.2) | 0.010 |
| NEWS score, median, IQR[1] | 8 (6-10) | 9 (6 - 11.25) | 7 (6-10) | 0.039 |
| NEWS score > 5, n (%)[1] | 132 (89.2) | 37 (97.4) | 95 (86.4) | 0.073 |
| AST, U/L, median, IQR[1] | 63.7 (39.8-118.6) | 124.20 (59.71-285.98) | 56.95 (37.19-95.42) | <0.001 |
| AST > 120U/L[1] | 36 (24.32) | 19 (50) | 17 (15.5) | <0.001 |
| ALT, U/L, median, IQR[1] | 33.7 (20.5-52.5) | 41.52 (25.15- 162.41) | 32.70 (19.0- 44.78) | 0.005 |
| ALT > 120U/L[1] | 18 (12.2) | 10 (26.3) | 8 (7.3) | 0.002 |
| PCT[1] | 3 (1-19) | 8.0 (1.0-26.75) | 2.0 (1.0-13.0) | 0.268 |
| CRP[1] | 131.5 (68.5-160.5) | 136.00 (67-162) | 128.90 (68.50-159.90) | 0.701 |

[1]Day 1 measurements; DIC: disseminated intravascular coagulation, CNS: central nervous system, HTN: hypertension DM: diabetes mellitus, IQR: inter-quartile range, ICU: intensive care unit, PLT: platelet, PT: prothrombin time, APTT: activated partial thromboplastine time, UVA: universal vital assessment, MEWS: modified early warning score, NEWS: national early warning score, AST: aspartate aminotransferase, ALT: alanine aminotransferase, PCT: procalcitonin, CRP: C-reactive protein

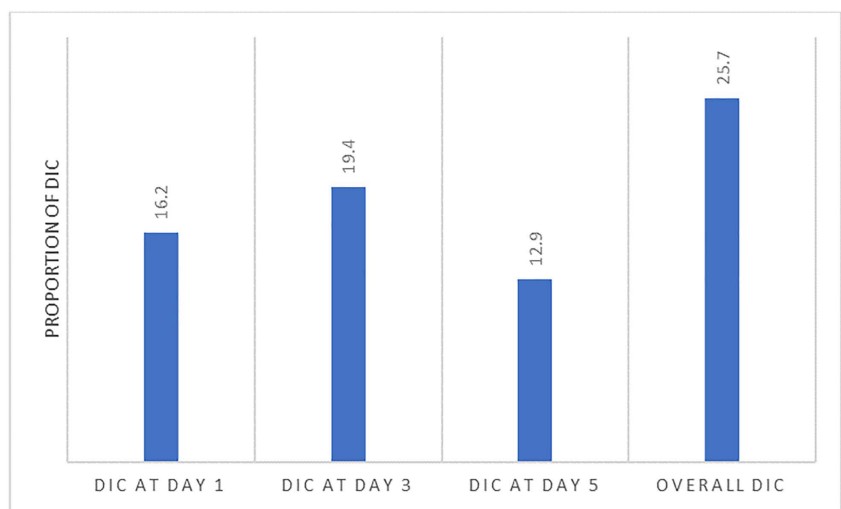

**Fig 2. Prevalence of DIC on different days after admission.** DIC; disseminated intravascular coagulation. DIC: Disseminated intravascular coagulation.

**Table 2. Bivariate and multivariate analyses of the associations between variables and DIC development in sepsis patients at JUMC from October 1, 2023, to September 30, 2024.**

| Variables | DIC Positive, n=38 | DIC negative, n=110 | COR (95% CI) | p value | AOR (95% CI) | P value |
|---|---|---|---|---|---|---|
| Sex, Male, n (%) | 20 (52.6) | 62 (56.4) | 1.16 (0.56-2.44) | 0.690 | | |
| Age<60 years, n (%) | 34 (89.5) | 98 (89.1) | 0.68 (0.24-1.96) | 0.478 | | |
| In ICU stay (days), median (IQR) | 3 (1–5) | 3 (2-4.25) | 0.99 (0.84-1.19) | 0.983 | | |
| **Interventions, n (%)** | | | | | | |
| Mechanical ventilation | 31 (81.6) | 72 (65.5) | 2.34 0.94-5.80 | **0.067** | 1.37(0.37-5.12) | 0.640 |
| Vasopressor use | 11 (28.9) | 25 (22.7) | 0.72 (0.32-1.66) | 0.442 | | |
| Antibiotic use | 22 (57.9) | 72 (65.50 | 0.77 (0.36-1.66) | 0.512 | | |
| **Site of infection, n (%)** | | | | | | |
| Lung | 16(42.1) | 56 (51) | 1.45 (0.69-3.06) | 0.326 | | |
| Abdomen | 7 (18.4) | 19 (19.3) | 0.94 (0.36-2.44) | 0.890 | | |
| Soft tissue | 5 (13.2) | 6 (5.5) | 0.49 (0.13-1.86) | 0.298 | | |
| CNS | 4 (10.5) | 9 (8.2) | 1.31 (0.38-4.52) | 0.672 | | |
| **Comorbidities, n (%)** | | | | | | |
| Respiratory disease | 20 (52.6) | 39 (35.5) | 0.49 (0.23-1.04) | **0.065** | 2.02 (0.79-5.11) | 0.139 |
| Heart disease | 2 (5.3) | 10 (9.10 | 1.80 (0.38-8.61) | 0.462 | | |
| Malaria | 11 (30) | 18 (16.4) | 0.55(0.23-1.32) | **0.181** | 1.81 (0.59-5.47) | 0.295 |
| Cancer | 1 (2.6)) | 4 (3.6)) | 1.39 (0.15-12.89) | 0.769 | | |
| Metabolic | 1 (2.6) | 19 (17.3) | 7.73 (0.99-59.82) | **0.050** | 2.64 (0.28-24.86) | 0.396 |
| **Organ failure, n (%)** | | | | | | |
| Liver failure | 2 (5.3) | 3 (2.7) | 0.51 (0.08-3.14) | 0.464 | | |
| Kidney failure | 2 (5.3) | 13 (11.8) | 2.41 (0.52-11.22) | 0.261 | | |
| Lung failure | 16 (42.1) | 39 (35.5) | 0.76 (0.36-1.60) | 0.465 | | |
| Brain failure | 4 (10.5) | 12 (11) | 1.04 (0.31-3.45) | 0.948 | | |
| **Laboratory results** | | | | | | |
| Positive blood culture, n (%) | 6 (15.8) | 9 (8.2) | 0.48 (0.16-1.44) | **0.188** | 2.978 (0.74-12.03) | 0.126 |
| AST>120U/L, n(%) | 19 (50) | 17 (15.5) | 5.65 (2.47-12.92) | **<0.001** | 4.39 (1.75-11.01) | **0.002** |
| ALT>120U/L, n (%) | 10 (26.3) | 8 (7.3) | 4.55 (1.64-12.62) | **0.004** | 1.49 (0.30-7.41) | 0.625 |
| PCT, median (IQR) | 8.0 (1.0-26.75) | 2.0 (1.0-13.0) | 0.99 (0.97-1.01) | 0.303 | | |
| CRP, median (IQR) | 136.0 (67-162) | 128.9(68.5-159.9) | 0.99 (0.99-1.01) | 0.695 | | |
| Leukocytosis, n (%) | 19 (50.0) | 66 (60.0) | | | | |
| Anemia, n (%) | 15 (45.5) | 48 (49.0) | | | | |
| Thrombocytopenia,n (%) | 26 (68.42) | 20 (18.20) | 5.73 (2.54-12.89) | **<0.001** | 6.04 (2.41-15.12) | **<0.001** |
| Prolonged PT, n (%) | 21 (55.26) | 33 (30) | 2.88 (1.35-6.15) | **0.006** | 3.40 (1.36-8.51) | **0.009** |
| Prolonged APTT, n (%) | 20 (52.63) | 31 (28.2) | 2.83 (1.32-6.06) | | 0.98 (0.30-3.16) | 0.972 |
| **Clinical scores, n (%)** | | | | | | |
| MEWS >5 | 31 (81.6) | 64 (58.2) | 3.18 (1.29-7.86) | **0.012** | 1.37 (0.40-4.62) | 0.615 |
| NEWS >5 | 9 (6 - 11.25) | 7 (6–10) | 5.45 (0.69-42.95) | **0.107** | 2.67 (0.27-26.87) | 0.404 |
| UVA>5 | 11 (28.9) | 38 (34.5) | 0.77 (0.35-1.72) | 0.528 | | |
| SIRS >2 | 24 (63.2) | 49 (44.5) | 2.13 (0.99-4.56) | **0.050** | 2.05 (0.82-5.16) | 0.127 |

ICU: intensive care unit, CNS: central nervous system, SIRS: systemic inflammatory response syndrome, PT: prothrombin time, APTT: activated partial thromboplastine time, UVA: universal vital assessment, MEWS: modified early warning score, NEWS: national early warning score, AST: aspartate aminotransferase, ALT: alanine aminotransferase, PCT: procalcitonin, CRP: c-reactive protein.

### Risk factors for ICU mortality

Length of stay in the ICU, procalcitonin level, creatinine level, SIRS score, UVA score and MEWS were the variables that were significantly associated with mortality. Multivariate analysis of the candidate variables revealed that there was no statistically significant association between the selected candidate variables and mortality in DIC patients (Table 3).

### Survival analysis

Based on the Kaplan-Meier survival analysis, the median in-ICU survival times of patients with and without DIC were 6 days and 13 days, respectively. The log-rank test revealed that there was no statistically significant difference in survival between patients with and without DIC (p < 0.328) (Fig 3).

### DIC score in predicting ICU mortality

The area under the curve (AUC) value for the JAAM score at ICU admission for the prediction of ICU mortality in sepsis patients with DIC was 0.787. A cutoff value of ≥5 yielded a sensitivity of 70.6% and a specificity of 71.4% (95% CI: 0.624–0.950, p = 0.003) (Fig 4 and Table 4).

**Table 3. Bivariate and multivariate analyses of the associations between variables and clinical outcomes in DIC-related septic patients at JUMC from October 1, 2023, to September 30, 2024.**

| Variables | Survivors, n = 21 | Non-survivors N = 17 | COR (95% CI) | p value | AOR (95% CI) | p value |
|---|---|---|---|---|---|---|
| Male, n (%) | 10 (47.6) | 10 (58.8) | 0.64 (0.18-2.31) | 0.493 | | |
| Age < 60years, n (%) | 17 (81.0) | 16 (94.1) | 0.38 (0.04-3.98) | 0.416 | | |
| In ICU stay (days), median (IQR) | 3 (1.5-6.0) | 2 (1-3.5) | 1.35 (0.96-1.89) | **0.081** | 1.49 (0.96-2.32) | 0.074 |
| Organ failure, n (%) | 13 (61.9) | 11 (64.5) | 0.89 (0.24-3.35) | 0.859 | | |
| Comorbidities, n (%) | 15 (71.4) | 12 (70.6) | 1.04 (0.26-4.26) | 0.955 | | |
| Positive blood culture, n (%) | 4 (19) | 2 (11.8) | 1.77 (0.28-11.04) | 0.544 | | |
| Thrombocytopenia, n (%) | 13 (61.9) | 13 (76.5) | 2.00 (0.48-8.32) | 0.341 | | |
| Prolonged PT, n (%) | 10 (47.6) | 11 (64.7) | 2.02 (0.54-7.49) | 0.295 | | |
| Prolonged APTT | 10 (47.6) | 10 (58.8) | 1.57 (0.43-5.71) | 0.493 | | |
| CRP, median (IQR) | 114 (60.3-158.8) | 144 (103.3-164) | 0.99 (0.98-1.01) | 0.296 | | |
| Procalcitonin, median (IQR) | 2 (0.9-18.9) | 14 (2.0-46.0) | 0.97 (0.94-1.01) | **0.093** | 0.98 (0.94-1.02) | 0.341 |
| Creatinine, median (IQR) | 0.91 (0.76-1.37) | 0.95 (0.77-2.17) | 0.62 (0.31-1.23) | **0.171** | 0.48 (0.22-1.02) | 0.056 |
| AST > 120 U/L, n (%) | 10 (47.6) | 9 (56.2) | 1.41 (0.38-5.23) | 0.603 | | |
| ALT > 120 U/L, n (%) | 5 (23.8) | 5 (29.4) | 0.75 (0.18-3.19) | 0.697 | | |
| SIRS >2, n (%) | 12 (57.1) | 12 (70.6) | 0.62 (0.28-1.35) | **0.228** | 2.13 (0.59-7.63) | 0.244 |
| UVA > 5, n (%) | 5 (23.8) | 6 (35.6) | 0.84 (0.65-1.09) | **0.195** | 1.25 (0.75-2.07) | 0.393 |
| MEWS >5, n (%) | 16 (76.2) | 15 (88.2) | 0.74 (0.53-1.03) | **0.074** | 0.52 (0.248-1.07) | 0.077 |
| NEWS >5, n (%) | 20 (95.2) | 17 (100) | 0.97 (0.78-1.19) | 0.762 | | |

ICU; intensive care unit, PT; prothrombin time, APTT; activated partial thromboplastin time, CRP; C-reactive protein, AST; aspartate aminotransferase, ALT; alanine aminotransferase, SIRS; systemic inflammatory response syndrome, UVA; universal assessment score, NEWS; national early warning score.

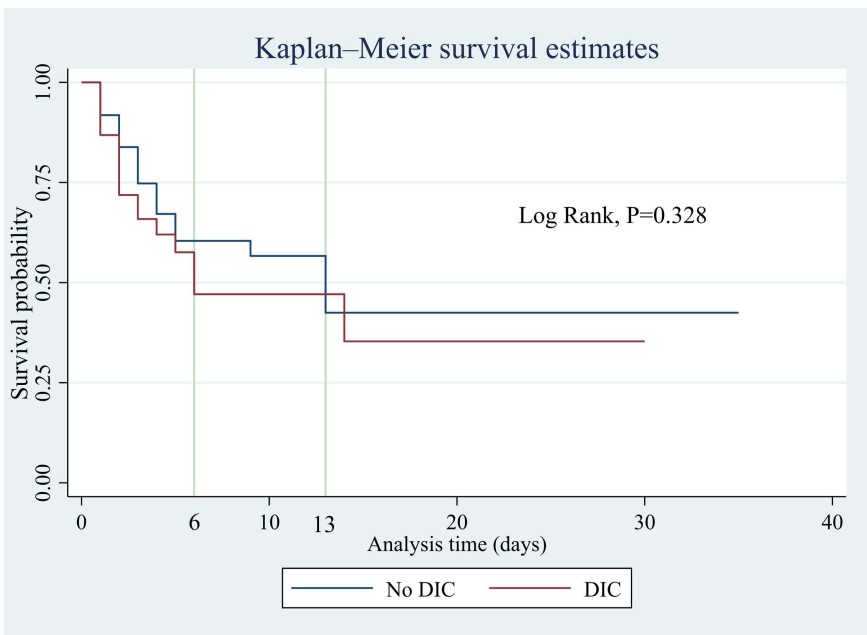

**Fig 3. Kaplan–Meier survival analysis according to the presence of DIC in sepsis patients. DIC: Disseminated intravascular coagulation.**

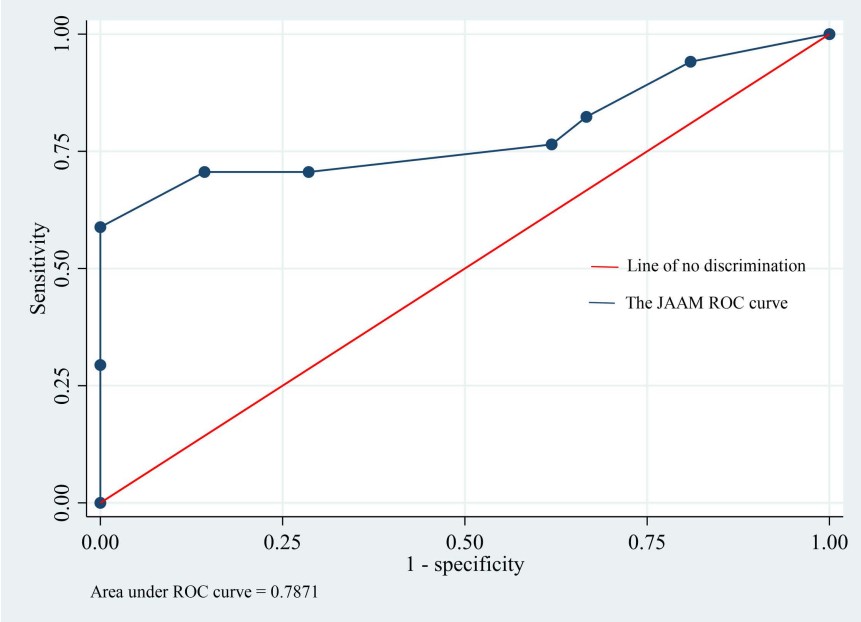

**Fig 4. Receiver operating characteristic analysis of the ability of the JAAM score to predict mortality in sepsis patients with DIC.**

**Table 4. Sensitivity and specificity of the JAAM score in predicting in-ICU mortality in sepsis patients with DIC.**

| Area Under the Curve | | | | | | | |
|---|---|---|---|---|---|---|---|
| Test Result Variable(s): JAAM score at day1 | | | | | | | |
| AUC | Cutoff value | Sensitivity | Specificity | Std. Error[a] | Asymptotic Sig.[b] | Asymptotic 95% Confidence Interval | |
| | | | | | | Lower Bound | Upper Bound |
| 0.787 | ≥5 | 70.6 | 71.4 | 0.083 | 0.003 | 0.624 | 0.950 |

## Discussion

Disseminated intravascular coagulation is a common complication and is associated with poor prognosis in sepsis patients admitted to the ICU. In the present study, the overall prevalence of DIC in sepsis patients in the ICU was 25.7%, varying on different days after admission, with values of 16.2%, 19.4%, and 12.9% on days 1, 3 and 5 of admission, respectively. Increased AST, thrombocytopenia, and prolonged PT were found to be independent predictors of DIC in sepsis patients. While the in-ICU mortality rates were not significantly different between patients with and without DIC, the median in-ICU survival were reported to be 6 days and 13 days, respectively. Moreover, the JAAM score was found to predict mortality with an AUC of 0.787, and at a cutoff point of ≥ 7, the sensitivity and specificity to predict mortality were 100% and 75%, respectively.

Previous studies that used the JAAM criteria to diagnose DIC reported a higher prevalence of DIC in sepsis patients than did the current study. Seven studies from Japan reported that the prevalence of DIC ranged from 29% to 91% [5,27–32]. However, in the studies that reported very high prevalence rates, participants were patients with coagulopathy with antithrombin activity less than 70% and were treated with antithrombin [28] or recombinant thrombomodulin [27]. A retrospective study from China involving sepsis patients with combined coagulation abnormalities reported a prevalence of 45% [33]. A study from South Korea involving adult patients with severe sepsis or septic shock reported a 55% DIC prevalence [34]. A study in France that recruited adults with septic shock reported a DIC prevalence of 34% [35]. Overall, a meta-analysis involving 11 studies that reported positive JAAM scores in sepsis patients reported a pooled prevalence of 55% [36]. The variation in the magnitude of DIC across studies is attributed to differences in the patient population, with some patients experiencing septic shock and others having known coagulopathy.

The present study reported a higher prevalence of DIC (19.4%) on day 3 after admission, which is not consistent with previous studies that reported a higher prevalence on the day of admission than on the subsequent days after admission [28,35]. The variation in disease severity on the day of admission might have created this difference. The mortality rates in patients with and without DIC were 44.7% and 33.6%, respectively. Survival analysis revealed that there was no statistically significant difference between patients with and without DIC (P<0.328). Previous studies reported mortality rates ranging from 22% to 61% among sepsis patients with DIC [5,27–30,33,35]. The present study further demonstrated that the JAAM DIC score on the day of admission predicted ICU mortality in sepsis patients with DIC, with an AUC of 0.787 (95% CI: 0.624–0.950, p=0.003). At a cutoff point of 5 or greater, the JAAM score can predict mortality with 70.5% sensitivity and 71.4% specificity. A previous study involving adults with severe sepsis and septic shock reported that the JAAM score predicted mortality, with an AUC of 0.717 [31]. This finding was different from that of a study conducted in Korea on the performance of five different DIC diagnostic criteria for predicting mortality in patients with complicated sepsis, which revealed that the JAAM DIC score predicted ICU mortality with an AUC of 0.616 (95% CI: 0.498–0.733, p=0.065) [34].

In this study, an AST > 120 was associated with DIC development in sepsis patients, with an odds ratio of 4.39 (95% CI: 1.75–11.01). AST is an enzyme test that measures the amount of aspartate aminotransferase in the blood, which helps us assess liver health. An increased level of AST could indicate liver dysfunction. Liver dysfunction often occurs in sepsis due to pathogens, toxins or inflammatory mediators [37]. Sepsis-associated liver dysfunction (SALD) is also reported in

sepsis which includes hepatocellular injury, cholestatic injury, and shock liver [38]. Secretion of inflammatory cytokines like interleukin (IL)-6 during sepsis leads to the production of acute-phase proteins such as C-reactive proteins, which in turn results in inhibition of the protein C pathway, and thus increases coagulation activity [39]. Cytokines released from Kupffer cells and neutrophils such as tumor necrosis alpha (TNF-α), IL-1β, IL-6, IL-12, and IL-18, reactive oxygen species (ROS), and nitric oxide (NO) are responsible for damage of endothelial cells and hepatocytes [40]. Liver impairment disrupts the capacity of the liver to effectively manage metabolic and detoxification processes, critically undermining the management and prognosis of sepsis [41]. Pathophysiologically, liver disease is a risk factor for DIC in septic patients because the liver produces factors that play a role in coagulation [38]. In liver disease, there is a decrease in the production of regulatory antithrombin, protein C, or protein S [39]. These changes lead to an imbalance between coagulation and fibrinolysis processes, representing potential risk factors for developing DIC [22]. Additionally, since DIC is characterized by widespread activation of the coagulation cascade, it can signal underlying systemic inflammation that damages various tissues including the liver and muscles, resulting in elevated AST levels.

Another factor associated with the development of DIC in sepsis patients is thrombocytopenia, with an odds ratio of 6.04 (95% CI: 2.41–15.12). Thrombocytopenia is a relatively common complication in patients with sepsis. In sepsis-associated DIC, marked coagulation activation causes multiple microfibrin thrombi within the systemic microvasculature. As the condition progresses, platelets are consumed along with coagulation factors, resulting in thrombocytopenia [40]. Numerous studies have shown that thrombocytopenia in sepsis patients is linked to poor prognosis, including prolonged hospital stays, lower survival rates, shock, bleeding, acute kidney injury [42] and hemostatic abnormalities ranging from subclinical coagulopathy to DIC.

Immunothrombosis is the primary underlying cause of thrombocytopenia in patients with infectious diseases and plays a physiological role in preventing systemic pathogen dissemination [43]. Increased platelet consumption and reduced platelet production are likely attributable to a systemic inflammatory response with severe infections, leading to sepsis-associated thrombocytopenia [42], which is one of the diagnostic criteria of DIC. During infection by bacterial endotoxins, especially gram-negative infections, both the coagulation and fibrinolytic systems are known to be strongly activated, leading to platelet activation and systemic thrombosis [44]. Thus, changes in the PLT can influence the scoring of DIC.

We also reported that prolonged prothrombin time (PT) is associated with DIC, with an odds ratio of 3.40 (95% CI: 1.36–8.51). Prolonged PT refers to an increased time for blood to clot in a laboratory test that measures the extrinsic and common pathway of coagulation, which could be due to impaired synthesis and increased consumption of clotting factors. DIC is a coagulopathy characterized by the consumption of clotting factors and platelets [1]. In sepsis, many inflammatory cytokines are produced and released into the circulation, leading to excessive activation of the coagulation process, fibrinolysis impairment, and suppression of anticoagulant mechanisms [11]. The impairment of fibrinolysis contributes to the hypercoagulable state due to excessive production of PAI 1. This potentially leads to a prothrombotic state and organ dysfunction due to tissue hypoperfusion. This results in the prolongation of PT and thrombocytopenia [44].

Furthermore, we analyzed factors associated with clinical outcomes in terms of the ICU mortality of sepsis patients that developed DIC. According to the multivariate analysis, no variable was found to be associated with ICU mortality. The variables found to be significant in the bivariate analysis were the length of stay in the ICU, procalcitonin level, creatinine level, SIRS score, UVA score and MEWS score. The limitation of this research was that it was conducted in a single center, requiring precaution when inferring the findings to the broader population beyond the study setting. Sepsis-3 definition which employs SOFA score for sepsis diagnosis was not used due to lack of all components. This might limit the comparability of the study with those that used sepsis-3 to define sepsis.

## Conclusions

A quarter of ICU-admitted septic adults developed DIC. The magnitude was greater on the 3rd day of admission, highlighting the need for close follow-up of these patients for DIC development. The liver enzymes AST, thrombocytopenia and prolonged

PT are associated with DIC development. A change in these parameters could help in further examining patients for DIC. There was no significant difference in mortality with respect to the presence of DIC in sepsis patients. The JAAM score used to diagnose DIC in sepsis patients can be used as a predictor of ICU mortality among septic adults with DIC.

## Supporting information

**S1 Table. Diagnostic criteria of DIC and their scoring.**
(DOCX)

**S1 Dataset. Dataset used for all analysis.**
(XLSX)

## Author contributions

**Conceptualization:** Girum Tesfaye Kiya.

**Data curation:** Girum Tesfaye Kiya, Gedion Milkias, Iyasu Demeke, Aragaw Fiseha.

**Formal analysis:** Girum Tesfaye Kiya, Gedion Milkias.

**Funding acquisition:** Girum Tesfaye Kiya, Zeleke Mekonnen, Elsah Tegene Asefa, Gedion Milkias, Edosa Tadasa, Edosa Kejela, Iyasu Demeke, Aragaw Fiseha, Gemeda Abebe.

**Investigation:** Girum Tesfaye Kiya, Gedion Milkias, Edosa Tadasa, Edosa Kejela, Gemeda Abebe.

**Methodology:** Girum Tesfaye Kiya, Zeleke Mekonnen, Elsah Tegene Asefa, Gedion Milkias, Edosa Tadasa, Edosa Kejela, Gemeda Abebe.

**Project administration:** Girum Tesfaye Kiya, Elsah Tegene Asefa, Gemeda Abebe.

**Resources:** Girum Tesfaye Kiya.

**Software:** Girum Tesfaye Kiya.

**Supervision:** Zeleke Mekonnen, Elsah Tegene Asefa, Edosa Tadasa, Gemeda Abebe.

**Validation:** Girum Tesfaye Kiya, Zeleke Mekonnen, Elsah Tegene Asefa, Edosa Kejela, Iyasu Demeke, Aragaw Fiseha, Gemeda Abebe.

**Visualization:** Girum Tesfaye Kiya.

**Writing – original draft:** Girum Tesfaye Kiya, Gedion Milkias.

**Writing – review & editing:** Zeleke Mekonnen, Elsah Tegene Asefa, Edosa Tadasa, Edosa Kejela, Iyasu Demeke, Aragaw Fiseha, Gemeda Abebe.

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
