## [Decision Letter · Decision Letter 0]

13 Jun 2025

Dear Dr. Kiya,

Thank you for submitting your manuscript to PLOS ONE. After careful consideration, we feel that it has merit but does not fully meet PLOS ONE’s publication criteria as it currently stands. Therefore, we invite you to submit a revised version of the manuscript that addresses the points raised during the review process.

We look forward to receiving your revised manuscript.

Kind regards,

Kovuri Umadevi

Academic Editor

PLOS ONE

Journal Requirements:

2. Please include captions for your Supporting Information files at the end of your manuscript, and update any in-text citations to match accordingly. Please see our Supporting Information guidelines for more information: http://journals.plos.org/plosone/s/supporting-information .

Additional Editor Comments:

Subject: Decision on Manuscript PONE-D-25-18967 – Major Revision Required

Dear Dr. Kiya,

We have completed the peer review process for your manuscript titled “Disseminated intravascular coagulation, associated factors and clinical outcomes among critically ill septic adults admitted to a tertiary hospital in Ethiopia: A prospective longitudinal study” (Manuscript Number: PONE-D-25-18967).

Please note that one of the reviewers has recommended rejection. However, the editorial decision at this stage is Major Revision.

This decision does not guarantee acceptance. You are requested to carefully address all reviewer comments in your revised manuscript. The final decision will depend on how well the revision satisfies the concerns raised, especially those of the reviewer who recommended rejection.

We encourage you to provide a detailed point-by-point response along with your revised submission.

Thank you,

Dr. Kovuri Umadevi

Reviewers' comments:

Reviewer's Responses to Questions

**Comments to the Author**

1. Is the manuscript technically sound, and do the data support the conclusions?

Reviewer #1: Partly

Reviewer #2: Partly

2. Has the statistical analysis been performed appropriately and rigorously?

Reviewer #1: I Don't Know

Reviewer #2: Yes

3. Have the authors made all data underlying the findings in their manuscript fully available?

Reviewer #1: Yes

Reviewer #2: Yes

4. Is the manuscript presented in an intelligible fashion and written in standard English?

Reviewer #1: Yes

Reviewer #2: Yes

Reviewer #1: How were the patients defined/diagnosed as "septic"? I suggest to the authors the inclusion of the reference.

Also, how were the patients diagnosed as "serious liver disorders"?

How were the patients diagnosed as kidney failure, brain failure, lung failure and liver failure?

Reviewer #2: General Comments:

This is a prospective cohort study based on ICU patients from a local hospital, enrolling a total of 148 patients diagnosed with sepsis. The manuscript is generally well-structured, employs appropriate statistical methods, and contributes to the understanding of DIC development among septic patients within the context of the local disease spectrum. However, there are concerns regarding the rigor of the discussion, the originality of the conclusions, and the presentation quality of several figures. Overall, I believe that the manuscript does not yet meet the publication standards of this journal. Nevertheless, I am confident that, with careful revision, particularly in improving the clarity of visualizations and the scientific robustness of the conclusions, the manuscript could be significantly enhanced.

Specific Comments:

1. Figure Quality and Presentation:

The flowchart in Figure 1 could be visually improved. Many online tools are available to create streamlined, reader-friendly diagrams that would enhance the visual clarity of your results. Similar issues are noted in Figure 2 and Figure 4. In Figure 2, the bar plot would better reflect the differences in patient numbers if the bars were left-aligned rather than center-aligned. In Figure 4, the AUROC plot lacks a legend. While it can be inferred that the red line may represent a random classifier, this should be clearly labeled within the figure, and AUROC values should be explicitly reported.

2. Discussion of AST and DIC Association:

The discussion on the association between AST levels and the occurrence of DIC in ICU septic patients is less rigorous compared to the discussion of other factors. First, the authors should elaborate on the changes in AST levels that may result from sepsis, similar to their explanation of sepsis-induced thrombocytopenia and prolonged PT. Second, the current argument linking AST to DIC mainly relies on the intermediary presence of liver disease, without direct evidence connecting AST elevation to DIC. Given that AST elevation can be influenced by various non-hepatic conditions, and that liver disease may lead to a wide range of complications beyond coagulopathy, the causal inference here appears somewhat tenuous. A more comprehensive literature review and clearer mechanistic explanation are warranted.

3. Clarification on Sepsis Definition:

A 2016 article published in Critical Care entitled "Revision of the Japanese Association for Acute Medicine (JAAM) disseminated intravascular coagulation (DIC) diagnostic criteria using antithrombin activity" discussed an update of the JAAM criteria to align with the Sepsis-3 definition. While the JAAM scoring system is a key variable in this study, the supplementary material indicates that the Sepsis-3 definition was not adopted. The authors should clarify the rationale behind not using the Sepsis-3 criteria—whether due to data limitations, clinical considerations, or other constraints.

4. Interpretation of JAAM Score Cut-off:

The manuscript suggests that a JAAM score ≥7 can predict mortality in ICU septic patients with DIC. However, given that the JAAM scoring range is 0–8, and that a score ≥4 already meets the diagnostic threshold for DIC, the clinical utility of using such a high cutoff (≥7) as a prognostic marker may be limited. A score this high suggests an already advanced coagulopathy and poor prognosis, potentially reducing the incremental predictive value of the finding. Moreover, previous studies have already demonstrated the prognostic relevance of the JAAM score in severe sepsis. Thus, the novelty of this conclusion appears limited and should be further discussed.

5. Ambiguity in the Conclusion Statement:

The final sentence of the conclusion is somewhat ambiguous. It is unclear whether the authors propose that the JAAM score can serve as a prognostic marker for all ICU patients, or specifically for septic ICU patients with DIC. This should be clearly stated in both the discussion and conclusion sections to avoid misinterpretation.

**Do you want your identity to be public for this peer review?** For information about this choice, including consent withdrawal, please see our Privacy Policy

Reviewer #1: **Yes: ** Felipe Martins Liporaci

Reviewer #2: No

---

## [Author Response · Author response to Decision Letter 1]

26 Jun 2025

Response to reviewers

Dear Editor and reviewers,

We are grateful for the valuable and constructive comments provided. Your critical evaluation of the manuscript will definitely improve the work. We made a point-by-point response to your comments and suggestions as follows, and changes were made in the manuscript using track changes:

Reviewer #1:

1. How were the patients defined/diagnosed as "septic"? I suggest to the authors the inclusion of the reference.

Response: Thank you for the critical inquiry regarding the definition used for sepsis diagnosis. As it is indicated in the limitation of the study, the Sepsis-3 definition, which employs the SOFA score for sepsis screening, was not used in this study. The practicality of using the SOFA score is highly limited, particularly in resource-constrained settings like ours. Additionally, the development of the SOFA score did not take into account data from low-resource countries. The Surviving Sepsis Campaign (SSC) 2021 guideline recommends the use of “a performance improvement program for sepsis,” which includes sepsis screening for the acutely ill. However, there is no single screening tool that wins universal assent due to the heterogeneity of the disease and variability of settings across the world.

Recognizing these challenges, the SSC guidelines encourage hospitals to adopt the most accurate, feasible, and sustainable approach to sepsis screening. Although qSOFA was proposed as a simplified screening tool, previous studies indicated that it is not suitable for ICU settings. Consequently, many ICUs continue to rely on the SIRS score and physicians’ clinical judgment based on readily available parameters. Likewise, in our study, sepsis diagnoses were determined by attending physicians, taking into account clinical, laboratory, and radiological findings. Clinical investigations included pulse rate, respiratory rate, temperature, oxygen saturation, and Glasgow Coma Scale, while the laboratory analyses included culture, CBC, procalcitonin, C-reactive protein, and organ function tests.

A subsection in the method part is added to describe the sepsis diagnosis used in this study. A reference (1) has also been cited based on the reviewer's request. Kindly refer to this reference in the last section (section 9: Screening), page 27, recommendation 1.

2. Also, how were the patients diagnosed as "serious liver disorders"?

Response: Thank you for the concern raised regarding serious liver disorder diagnosis. We hope that this question arises from the statement in the exclusion criteria. We are sorry for the mistake we made in describing serious liver disorder as one of the exclusion criteria. We had to omit the phrase from the exclusion criteria. During protocol development, we planned to exclude patients with serious liver disorders to avoid confounding effects on the coagulopathy. However, later during research, we decided that excluding anticoagulant use and known coagulopathy would suffice. We now omitted serious liver disorder from the exclusion criteria.

3. How were the patients diagnosed as kidney failure, brain failure, lung failure and liver failure?

Response: Thank you for the valuable questions raised regarding the diagnosis of organ failures. The following criteria were used in the study area’s ICUs to diagnose organ failure:

Liver failure was diagnosed considering significantly deranged liver functional tests including albumin, bilirubin, and INR.

Kidney failure was diagnosed when the eGFR was less than 15 ml/min/1.73 m² and when there was a need for renal replacement therapy (RRT).

Respiratory failure was diagnosed when lungs couldn’t exchange O₂ and CO₂, leading to abnormal gas exchange. In addition to the respiratory rate >35 or <8/min, the pulse oximetry reading with a saturation less than 91% was considered as a surrogate for PaO2 < 60 mmHg.

Brain death/failure was diagnosed when there was a coma, absence of brain stem reflexes, and apnea (lack of spontaneous breathing).

1. Surviving Sepsis Campaign. Early Identification of Sepsis on the Hospital Floors: Insights for Implementation of the Hour-1 Bundle [Internet]. 2019 [cited 2024 Dec 23]. Available from: https://sccm.org/survivingsepsiscampaign/hour-1-bundle-implementation-guide

Reviewer #2:

General Comments:

This is a prospective cohort study based on ICU patients from a local hospital, enrolling a total of 148 patients diagnosed with sepsis. The manuscript is generally well-structured, employs appropriate statistical methods, and contributes to the understanding of DIC development among septic patients within the context of the local disease spectrum. However, there are concerns regarding the rigor of the discussion, the originality of the conclusions, and the presentation quality of several figures. Overall, I believe that the manuscript does not yet meet the publication standards of this journal. Nevertheless, I am confident that, with careful revision, particularly in improving the clarity of visualizations and the scientific robustness of the conclusions, the manuscript could be significantly enhanced.

Specific Comments:

1. Figure Quality and Presentation: The flowchart in Figure 1 could be visually improved. Many online tools are available to create streamlined, reader-friendly diagrams that would enhance the visual clarity of your results. Similar issues are noted in Figure 2 and Figure 4. In Figure 2, the bar plot would better reflect the differences in patient numbers if the bars were left-aligned rather than center-aligned. In Figure 4, the AUROC plot lacks a legend. While it can be inferred that the red line may represent a random classifier, this should be clearly labeled within the figure, and AUROC values should be explicitly reported.

Response: Thank you so much for the valuable comments provided on the clarity of the figures. We modified the figures (Fig 1, 2, and 4) to improve the clarity.

2. Discussion of AST and DIC Association: The discussion on the association between AST levels and the occurrence of DIC in ICU septic patients is less rigorous compared to the discussion of other factors. First, the authors should elaborate on the changes in AST levels that may result from sepsis, similar to their explanation of sepsis-induced thrombocytopenia and prolonged PT. Second, the current argument linking AST to DIC mainly relies on the intermediary presence of liver disease, without direct evidence connecting AST elevation to DIC. Given that AST elevation can be influenced by various non-hepatic conditions and that liver disease may lead to a wide range of complications beyond coagulopathy, the causal inference here appears somewhat tenuous. A more comprehensive literature review and clearer mechanistic explanation are warranted.

Response: Thank you for the insightful comments. We addressed the comments by describing the effect of sepsis on AST and the possible mechanisms underlying elevated AST in DIC beyond the liver disease. We added statements from line 304 to 311, and from line 317 to 320.

3. Clarification on Sepsis Definition:

A 2016 article published in Critical Care entitled "Revision of the Japanese Association for Acute Medicine (JAAM) disseminated intravascular coagulation (DIC) diagnostic criteria using antithrombin activity" discussed an update of the JAAM criteria to align with the Sepsis-3 definition. While the JAAM scoring system is a key variable in this study, the supplementary material indicates that the Sepsis-3 definition was not adopted. The authors should clarify the rationale behind not using the Sepsis-3 criteria—whether due to data limitations, clinical considerations, or other constraints.

Response: Thank you for the important comment on the updated JAAM scoring system. As it has been indicated in the limitation of the study, the sepsis-3 definition was not used in this study due to a lack of some SOFA components. Data on antithrombin activity was also lacking to use the updated JAAM score.

4. Interpretation of JAAM Score Cut-off:

The manuscript suggests that a JAAM score ≥7 can predict mortality in ICU septic patients with DIC. However, given that the JAAM scoring range is 0–8, and that a score ≥4 already meets the diagnostic threshold for DIC, the clinical utility of using such a high cutoff (≥7) as a prognostic marker may be limited. A score this high suggests an already advanced coagulopathy and poor prognosis, potentially reducing the incremental predictive value of the finding. Moreover, previous studies have already demonstrated the prognostic relevance of the JAAM score in severe sepsis. Thus, the novelty of this conclusion appears limited and should be further discussed.

Response: Thank you for the insightful comments on the predictive role of the JAAM score. After this comment, we chose to reduce the cut-off value to >5 with a sensitivity of 70.5% and specificity of 71.4% (lines #258, 259, 296, 297, and table 4). Though the prognostic performance of the JAAM score was reported in previous studies, its role in predicting in-ICU mortality is scarcely reported. Additionally, validating its relevance across various contexts could offer meaningful contributions to the existing body of literature.

5. Ambiguity in the Conclusion Statement:

The final sentence of the conclusion is somewhat ambiguous. It is unclear whether the authors propose that the JAAM score can serve as a prognostic marker for all ICU patients or specifically for septic ICU patients with DIC. This should be clearly stated in both the discussion and conclusion sections to avoid misinterpretation.

Response: Thank you for the valuable comment. The result, discussion, and conclusion regarding the predictive role of the JAAM score are now corrected, describing its usefulness only among sepsis patients with DIC (lines #258, 261, 263, 295, and 366).

---

## [Decision Letter · Decision Letter 1]

7 Aug 2025

Disseminated Intravascular Coagulation, Associated Factors and Clinical Outcomes among Critically Ill Septic Adults Admitted to a Tertiary Hospital in Ethiopia: A Prospective Longitudinal Study

PONE-D-25-18967R1

Dear Dr. Girum Tesfaye Kiya

We’re pleased to inform you that your manuscript has been judged scientifically suitable for publication and will be formally accepted for publication once it meets all outstanding technical requirements.

Kind regards,

Kovuri Umadevi

Academic Editor

PLOS ONE

Additional Editor Comments (optional):

Dear Girum Tesfaye Kiya,

We are pleased to inform you that we have now received the required reviewer feedback and completed the editorial evaluation of your revised manuscript titled:

"Disseminated Intravascular Coagulation, Associated Factors and Clinical Outcomes among Critically Ill Septic Adults Admitted to a Tertiary Hospital in Ethiopia: A Prospective Longitudinal Study"

(Manuscript ID: PONE-D-25-18967R1) submitted to PLOS ONE.

Based on the positive reviewer comments and final assessment, we are delighted to inform you that your manuscript has been accepted for publication.

Congratulations, and thank you for choosing PLOS ONE as the platform for your research.

Best regards,

Dr. Kovuri Umadevi

Academic Editor

PLOS ONE

Reviewers' comments:

Reviewer's Responses to Questions

**Comments to the Author**

Reviewer #1: All comments have been addressed

2. Is the manuscript technically sound, and do the data support the conclusions?

Reviewer #1: Yes

3. Has the statistical analysis been performed appropriately and rigorously?

Reviewer #1: I Don't Know

4. Have the authors made all data underlying the findings in their manuscript fully available?

Reviewer #1: Yes

5. Is the manuscript presented in an intelligible fashion and written in standard English?

Reviewer #1: Yes

Reviewer #1: (No Response)

**Do you want your identity to be public for this peer review?** For information about this choice, including consent withdrawal, please see our Privacy Policy

Reviewer #1: **Yes: ** Felipe Martins Liporaci

---

## [Editor Report · Acceptance letter]

PONE-D-25-18967R1

PLOS ONE

Dear Dr. Kiya,

I'm pleased to inform you that your manuscript has been deemed suitable for publication in PLOS ONE. Congratulations! Your manuscript is now being handed over to our production team.

Kind regards,

on behalf of

Dr. Kovuri Umadevi

Academic Editor

PLOS ONE